# Me, Myself and My Insula: An Oasis in the Forefront of Self-Consciousness

**DOI:** 10.3390/biology12040599

**Published:** 2023-04-14

**Authors:** Alice Tisserand, Nathalie Philippi, Anne Botzung, Frédéric Blanc

**Affiliations:** 1Geriatrics and Neurology Units, Research and Resources Memory Center (CMRR), Hôpitaux Universitaires de Strasbourg, 67000 Strasbourg, France; nathalie.philippi@chru-strasbourg.fr (N.P.); anne.botzung@chru-strasbourg.fr (A.B.); f.blanc@unistra.fr (F.B.); 2ICube Laboratory (CNRS, UMR 7357), 67000 Strasbourg, France

**Keywords:** insula, self, interoception, autobiographical memory

## Abstract

**Simple Summary:**

The insula, or the fifth lobe of the brain, is involved in a wide variety of functions, including processes related to the self. The self is a complex construct comprising both a subjective–objective dimension and a temporal dimension. However, because of the lack of direct investigation, it remains unclear the way the insula is involved in the different aspects of the self. This review describes the insula from both an anatomical and a functional point of view, the self through its different dimensions and the way the insula is involved in the self, relying on studies in healthy controls and in various affections. Understanding the link between the insula and the self might lead to improvement in care provision.

**Abstract:**

The insula is a multiconnected brain region that centralizes a wide range of information, from the most internal bodily states, such as interoception, to high-order processes, such as knowledge about oneself. Therefore, the insula would be a core region involved in the self networks. Over the past decades, the question of the self has been extensively explored, highlighting differences in the descriptions of the various components but also similarities in the global structure of the self. Indeed, most of the researchers consider that the self comprises a phenomenological part and a conceptual part, in the present moment or extending over time. However, the anatomical substrates of the self, and more specifically the link between the insula and the self, remain unclear. We conducted a narrative review to better understand the relationship between the insula and the self and how anatomical and functional damages to the insular cortex can impact the self in various conditions. Our work revealed that the insula is involved in the most primitive levels of the present self and could consequently impact the self extended in time, namely autobiographical memory. Across different pathologies, we propose that insular damage could engender a global collapse of the self.

## 1. Introduction

What makes us feel unique, living in the same body all life long, and what makes our life stories arise, is what we can call the self. The self has different dimensions, such as a physical one—the body as a whole or within its different parts—and a mental one—the first-person perspective of the self, existing and acting in the present time, and also knowledge of the self, including episodic and semantic memory knowledge. Different nomenclatures of the self exist, converging toward a hierarchical conception that ranges from the most internal bodily phenomena of awareness to very specific details of autobiographical memory. However, the anatomical substrates from which the self emerges in the brain remain unclear. A body of research suggests that a network of right frontoparietal structures is deeply involved in generating awareness, for instance, the right frontal and parietal lobes provide the ability to recognize the self-face, the self-body but also the self-voice [1,2]. These regions are therefore known to overlap with areas involved in the default-mode network, notably cortical midline structures that show activity during tasks requiring self-referential processing [3]. Whereas numerous brain substrates support personality and memories, one specific structure might host the very primitive levels of the self, namely the insula. The insula, or the fifth lobe of the brain, is folded deep inside the frontal, temporal and parietal opercula. It is a multiconnected brain region, with multiple roles, notably, on the one hand, a sensorimotor processing posterior part integrating the primary interoceptive cortex and, on the other hand, a supporting and integrating anterior part based on viscerosensory responses, socio-emotional processing and cognitive functions that allow the emergence of awareness.

In this review, we begin by describing the insula from an anatomical and functional point of view. We then present the main different cognitive models of the self, emphasizing the aspects related to the present self and overviewing the aspects related to the self extended in time, namely autobiographical memory. Finally, we will describe how the insula is topographically involved in the present self, and how insular damage can impact the self.

## 2. The Insula

### 2.1. General Anatomy and Functions of the Insula

Although still not well understood, the insula has many peculiarities, both functional and anatomical. Whereas some studies in rodents suggest that insular cortex (IC) organization is rather multimodal and shaped by learning [4,5], others have argued for a topographical organization with a functional complexity increasing along a posterior–anterior axis [6]. The insula is involved in a wide variety of functions ranging from sensory and affective processing to high-level cognition. The current thinking postulates the existence of four functionally distinct regions. This is illustrated in Figure 1, with (1) a sensorimotor region in the mid-posterior insula; (2) a central-olfactogustatory region; (3) a socio-emotional region in the anterior-ventral insula; and (4) a cognitive anterior-dorsal region [7]. From a structural point of view, one can distinguish three different areas of the IC, with a heterogeneous cytoarchitecture that ranges from granular in the posterior portion to agranular in the anterior portion. The posterior and dorsal part of the IC, whose structure is close to the parietal and temporal opercula, is described as “hypergranular”. The granularity progressively decreases from the postero-dorsal part to the antero-ventral part, toward an agranular cortex structure, with a predominant intermediate dysgranular part [8].

As reported by Craig, the posterior insular cortex (PIC) would rather be associated with sensorimotor processing, such as visceral sensations, autonomic control and interoception [9,10], whereas the anterior insular cortex (AIC) would support and integrate socio-emotional processing and cognitive functions [11,12,13,14]. A special feature of the AIC is the concentration of clusters of large pyramidal neurons: the von Economo neurons (VENs) [15], which are specific to mammals with well-developed socialization skills, such as hominoid primates, elephants, horses, pigs, cows and particularly certain cetaceans, such as the bowhead whale [16,17,18]. In addition to its intrinsically anatomical and functional complexity, the insula is a multiconnected brain region. Widespread connections are observed between both the right and left insula and other brain regions [19]. Regarding the frontal lobe, the insula has connections with the inferior, middle and superior gyri, the orbitofrontal cortex, the precentral gyrus, the frontal operculum and the subcallosal gyrus. Within the temporal lobe, the insula has connections with the superior temporal gyrus, including Heschl’s gyrus, the planum temporale, the planum polare, the temporal fusiform gyrus and the temporal operculum. With regard to the parietal lobe, the insula has connections with the supramarginal and angular gyri, the postcentral gyrus, the precuneus and superior parietal lobule and the parietal operculum. The insula also supports connections with the occipital lobe, notably the cuneus and lingual gyri and the occipital fusiform gyrus. Finally, the insula has connections with limbic areas, such as the thalamus, the amygdala, the hippocampus and parahippocampal gyrus, including the perirhinal and entorhinal cortices, the uncus and the posterior and anterior cingulate gyri, the latter comprising VENs. More precisely, there appear to be structural connectivity differences between the AIC and the PIC, and other brain regions. For instance, the AIC has a greater number of connections to anterior cortices (perigenual and subgenual anterior cingulate, anterior midcingulate, orbitofrontal, frontal and anterior temporal cortices), while the PIC has a greater number of connections to posterior cortices (dorsal posterior cingulate, posterior midcingulate, posterior temporal, sensorimotor, parietal and occipital cortices). However, both the anterior and the posterior part of the insula have connections with other limbic areas, such as the parahippocampal gyrus, including the entorhinal and perirhinal cortices, as well as the uncus [19,20,21,22,23]. Interestingly, the IC, and more precisely the AIC, is also anatomically and functionally connected to the infratentorial region of the brain, such as the brainstem, and particularly to specific nuclei: the nucleus tractus solitarii, the dorsal motor nucleus of the vagus, the parabrachial nuclei (sensory) and the periaqueductal gray (motor) [24,25,26,27]. The numerous connections found between the insula and other brain regions are consistent with the wide range of brain functions related to the insula. More broadly, insular connections with the frontal lobe have a role in language processes and in executive processes [28,29] involving an affective component, notably risk decision making [30]. Connections with auditory areas within the temporal lobe are related to the involvement of the IC in central auditory processing [31], whereas connections with parietal areas indicate a role in body scheme representations [32,33]. Insular-occipital connections, however, might be involved in emotional facial expression recognition [34]. Finally, the close relationship between the insula and the limbic system highlights the insula’s involvement in emotional processes; indeed, some researchers even consider that the insula constitutes an integral part of the paralimbic or limbic system [31]. The connections to the brainstem and diencephalon involved in interoception and homeostasis will be further discussed (see Section 2.2).

Given such diversity, both anatomical and functional, a question arises in the context of the self: through which processes can phenomena such as the interoceptive states of the body reveal socio-emotional states and give rise to such highly elaborated functions as awareness?

### 2.2. From Viscera to Insular Cortex

Among the preliminary work conducted on the role of the human insula was the seminal work by Wilder Penfield, using electrocortical stimulation, in the mid-20th century [10]. Penfield notably pointed out that stimulation of the PIC elicited a variety of visceral sensory—thus prompting researchers to dub the insula the “visceral brain”—and motor responses, as well as somatic sensory responses in the face, tongue and upper/lower limbs. Interestingly, stimulation of the insula unilaterally caused a sensation on the opposite side of the body but also sometimes on both sides of the body [35,36]. The insula is considered a primarily visceral-somatic region, but beyond visceral information processing, it has been proposed that the insula plays a broader role in interoception, i.e., the sense of the physiological condition of the body [37]. The interoceptive sensations arising from the body allow for a continuous monitoring of the state of the body through mechanisms, such as heart rate, blood pressure, respiration, proprioceptive signals and visceral activity [38,39]. Among the pioneering work on the functions of the insula, Penfield and Faulk found that stimulation of the lower part of the PIC produced abdominal sensations (e.g., “gurgling”, “rolling”, “pain”, “nausea” and “scratching”) and objective evidence of intra-abdominal motor activity (e.g., “borborygmus”, “belching” and “vomiting”), suggesting that the inferior PIC might monitor the sensory aspects of the stomach as well as gastric motility [35]. Nonetheless, to make the individual aware of all these sensations, the information concerning the internal state of the body is conveyed through a dedicated spinothalamocortical afferent system [40]. Interoceptive information travels within the nervous system in the form of a signal that is transmitted by several pathways and nuclei. A major component of this system consists of the Aδ and C fibers, whose role extends far beyond solely “pain and temperature” sensations. This system relates homeostatic information from all tissues of the body, innervates them and terminates monosynaptically in lamina I of the spinal and trigeminal dorsal horns. The signal therefore enters the spinal cord and trigeminal nucleus in the brainstem and conveys signals from the body structures of the head, the oral cavity, the skin of the face and scalp and the facial muscles of emotion and jaw movements to the posterior part of the ventromedial nucleus (VMpo) in the thalamus [41,42]. The VMpo anteriorly adjoins the basal ventromedial nucleus (VMb), which receives direct inputs from the nucleus of the solitary tract (NTS), within which parasympathetic afferents are carried (vagal and glossopharyngeal nerves). The VMpo and VMb then project to the PIC, progressing through the different portions of the insula. According to Craig and other researchers, Refs. [43,44], within the PIC, the interoceptive pathway produces a topographical representation of the body, from anterior to posterior aspects. The interoceptive signal is then integrated into the middle insular cortex (MIC), which has connections to the amygdala and hypothalamus, thus forming a combined representation of homeostatically salient features of the individual’s internal and external environment [34]. Direct stimulation of the MIC in the inferior part might be involved in a sense of unreality [36]. The AIC is reported to be an integrative site that represents “a common neural substrate for embodied and experiential processes” [45]. More specifically, the right AIC has been explicitly implicated in the mapping of the interoceptive state and response to heartbeat detection tasks, which are relevant tasks to assess visceral sensitivity [46]. The AIC constitutes a coordination site for “high-level homeostatic information, perhaps on the general state of the body, which is an important component of emotional experience and a sense of well-being” [47]. Picard et al. showed that electrical stimulation within the anterior-dorsal insula induces an intense feeling of bliss, involving both emotional and interoceptive components [13]. Thus, the AIC is involved in the representation of “cognitive feelings” which arises from the moment-to-moment integration of homeostatic information emanating from the body [40]. It has been further proposed that the AIC “instantiates all subjective feelings from the body and feelings of emotion” [6]. By generating an analogous “re-representation”, the AIC provides a basis for the subjective evaluation of the interoceptive state, which is therefore routed to the orbitofrontal cortex, a center of hedonism assessment. Finally, emotional experience and awareness occur as an emergent process across systems.

### 2.3. From Interoception to Awareness

Interoception is an active process that forwards neural information from the body to the brain and regulates vital processes at the most elementary level, while also modulating emotional experience. Bodily feelings are reported to be at the core of bodily awareness and even of the self [43,48,49]. Bodily awareness refers to the conscious perception of somatic and internal sensations and to awareness that these experiences are bound to the self [50,51,52,53,54]. It is a multidimensional construct that involves the feeling of owning a body and being the agent over one’s own actions [50,51,52].

As theorized by Craig [6,39], the sense of self results from a “cortical (that is, mental) integration of salience across all conditions”, at any moment in time, which constitutes homeostatic processes that determine what is salient to the individual. The foundation of Craig’s model is the perception of interoceptive neuronal signals as sensations, or “cognitive feelings” [39], through the creation of re-representations. Such signals generate pain, temperature, thirst, hunger, itch, muscle burn or ache, sensual touch, visceral urgency, flush and nausea, among other sensations [38]. Numerous studies have reported that the concept of the re-representation of the interoceptive condition of the body serves as a limbic sensory substrate for subjective feelings. Such studies show that activation in the right AIC and orbitofrontal cortices are associated with subjective emotion (e.g., recall-generated sadness, anger, anticipatory anxiety and pain, panic, disgust, sexual arousal, orgasm, trustworthiness and responses to music [55,56,57,58,59,60,61]) and that the right AIC is thus a fundamental structure for the generation of the mental image of one’s physical emotional state. The posterior-to-anterior anatomical model of integration from the primary interoceptive representations to the absolute representation of one’s feelings strongly suggests that awareness relies on homeostasis [40,62,63,64]. These data are consistent with Antonio Damasio’s hypothesis of the “somatic marker” which suggests that the subjective process of feeling emotions recruits brain regions that are involved in homeostasis. These feelings we perceive are the basis of our perceptions of ourselves, distinguishing between what belongs to the internal world and what belongs to the external world, and thus provide a neural basis to distinguish the self from the non-self. By integrating the interoceptive representation into a re-representation, the right AIC has the capacity to perceive the self as a physical and separate entity, in other words, subjective awareness. Thus, when first describing the intracerebral stimulation of the right insula, a patient might say, “I feel myself going” [35].

In the next section, we will focus on the different models of the self, before describing more precisely how the insula is involved in the different aspects of the self.

## 3. The Self

“I think, therefore I am”. Sense of self is an essentially human characteristic that provides one with the feelings of singularity, coherence, individuality and unity [65]. It is the mental process that unifies disparate experiences, levels of awareness, behaviors, cognitions and mental representations into a coherent, unified whole [66,67]. In common discourse, the term “self” refers to a feeling that something is “about me”. Reflecting on oneself requires that there is an “I” that can consider an object that is “me”. The question of the self has been widely explored by philosophers and psychologists. At the end of the 19th century, William James made an inventory of the physical self, mental self, spiritual self and the ego [62]. He distinguished the psychological process that is the subject of knowing and experiencing (the I-self) and the object of this awareness (the Me-self). His theory, heuristically of tremendous value, has dominated discussion around the sense of self and has been variously supplemented. For instance, Neisser suggested important distinctions between ecological, interpersonal, extended, private and conceptual aspects of the self [68]. These distinctions—or equivalents—reappear in more recent theories of the self as discussed in the neuroscience and neuropsychology fields, notably by researchers such as Antonio Damasio, Shaun Gallagher and Sally Prebble [48,65,69,70]. Different models emerged from their theories and hypotheses, but these present nuances rather than major differences (see Figure 2). As proposed by James in his pioneering work [62], more recent models support the idea that there is a phenomenological self (i.e., the I-self) and a conceptual self (i.e., the Me-self). The I-self refers to the subjective living experience that contributes to the construction of a mental representation of the self [71,72,73,74], whereas the Me-self supports the object of this representation, including knowledge about ourselves [74,75]. To complete William James’ work, most researchers have added a temporal dimension to their conception of the self that distinguishes the present self and the temporally extended self. The present self refers to the aspects of the sense of self that are related to and accessible in the present moment. As mentioned above, the temporally extended self refers to autobiographical memory and brings out the feeling of being the same person over time despite changes. In this review, we briefly overview the temporally extended self—or autobiographical memory—and we mainly focus on the present self, numerous aspects of which seem to be supported by the insula (see Section 3. Insula and the present self). The main theses are presented and discussed below. We focus particularly on Prebble’s model, which seems to be the most exhaustive, and we use its nomenclature to refer to the different components of the self. The authors approach the self along a subjectivity axis and a temporality axis, which give rise to four distinct components with different levels of awareness: the subjective sense of self (SSS), the self-concept (SC), the phenomenological continuity and the semantic continuity.

### 3.1. The Present Self 

On the one hand, the present self supports the SSS, which is the phenomenological part involving multiple levels of awareness. On the other hand, the present self supports the SC, which is the semantic part and involves the collection of knowledge about oneself [65].

#### 3.1.1. The Subjective Sense of Self

Prebble and collaborators also refer to the SSS as “the I-self”, as initially proposed by James [62]. The SSS constitutes the most elementary aspect of the sense of self. Humans would appear to share this component with most animals endowed with complex brains and sensory organs [6]. It is the component that confers the feeling of living in the present moment as a sentient being and is the cornerstone of the sense of self that allows the other components to exist and thereby makes possible the existence of autobiographical memory in humans. 

In Prebble and collaborators’ model, the SSS is made up of two hierarchically related forms of present-moment conscious self-experience: prereflective self-experience and self-awareness [65]. Prereflective experience refers to what some philosophers call “Qualia” and is related to an immediate and ongoing sensory or perceptual stream of experiences [69,76,77]. It can be understood as an integral feature of our conscious experience of the world [78,79]. In Neisser’s theory, the prereflective self-experience corresponds to the “ecological self” and the “interpersonal self”. The ecological self refers to self-body awareness and recognition, while the interpersonal self implies the feeling of agency. Damasio’s equivalent is the “protoself”—a pre-conscious state representing the first stage of the hierarchical process of awareness generation. The protoself refers to a collection of neural patterns that are representative of the body’s internal states. This aspect of the self constantly detects and records the internal physical changes that affect the homeostasis of the organism [80]. For Gallagher, the SSS corresponds to both a “minimal embodied self” and a “minimal experiential self”. The first one includes core biological and ecological aspects, which allow the system to distinguish between the self and the non-self. The second one contributes to an embodied sense of ownership (i.e., confers the feeling that I am the one undergoing the experience) and a sense of agency (i.e., confers the feeling that I am the one who is initiating or causing an action) [70]. Regarding the second component of the SSS in Prebble’s model, self-awareness reflects the ability to introspect about one’s mental states, behavior and experiences [81,82,83]. It represents a higher and reflective level of awareness that implies two important features. First, awareness is supposed to be directed inwardly, as opposed to toward the external environment [72,74,75,84,85,86]. Second, it is a reflective and meta-conscious experience that involves a capacity to observe, reflect, evaluate and focus attention on one’s subjective experience [72,74,75,82,87,88]. The equivalent is referred to as the “private self” in Neisser’s model and is based on awareness that the conscious experiences are exclusively our own. Damasio uses the expression “core consciousness” to refer to when the perception of the external world becomes conscious. He describes it as an emergent process that occurs when an organism becomes consciously aware of feelings associated with changes in internal bodily states [48]. Gallagher does not discuss this level of consciousness in his models. In the self-memory system (SMS) of Conway and Pleydell-Pierce, the authors consider the present self as a one and only component they call the “working self”, which is a sort of dynamic cognitive structure that acts as a central control process and modulates access to another component which is the long-term self.

The SSS, and especially the prereflective experience, is particularly difficult to evaluate. One way to measure it consists of assessing interoception, and some studies (see, infra, Section 4.1.1 and Section 4.1.2) have shown that it would be mainly based on the insula and on medial subcortical structures, such as the hypothalamus, which controls the general homeostasis of the organism, and the brain stem, whose nuclei map body signals [66]. Concerning self-awareness, a large range of neural regions are reported to be involved, including the following: cortical midline structures, such as the cingulate, medial prefrontal and parietal cortices; lateral cortical regions, such as the insula, the lateral prefrontal and parietal cortices and the temporal poles; and subcortical regions, notably the hypothalamus, the brainstem, the colliculi and the periaqueductal gray [89].

The SSS is a reflective process that is assumed to involve the construction of mental models of the self: the objective Me-self or SC.

#### 3.1.2. The Self-Concept

The SC is made up of autobiographical knowledge and refers to the way we internally represent who we are, determining what sort of person we are. According to Prebble’s model, the SC relies on all the attributes, traits, beliefs, values, social status, roles and physical characteristics we attribute to ourselves (e.g., I am a neuropsychologist, my hair is brown, I am French) but also contains self-esteem, self-image, goals, behavioral scripts and schemas and information about one’s material possessions and social relationships [62,65,68,90]. For most cognitive scientists, the SC is part of the present self but is highly dependent on autobiographical memory, which contributes to both the formation and maintenance of our knowledge about who we are [91,92,93,94,95,96,97]. In his 2013 model, Gallagher evokes some “psychological/cognitive aspects” of the self, which he describes as ranging “from explicit self-consciousness to conceptual understanding of self as self, to personality traits of which one may not be self-conscious at all”, but he also proposes to include the ability to represent oneself as oneself, while supporting the idea that psychological continuity and memory are important for personal identity [70]. Damasio’s theory does not isolate an SC component per se. In his view, there is an autobiographical self that emerges in extended consciousness, which he depicts as “a relatively stable collection of the unique facts that characterize a person” [98]. Conway, on the other hand, considers that the SC exists independently of specific, temporally defined incidents. However, in his model, he opted to integrate the SC in the long-term self because abstracted knowledge structures have to be connected to autobiographical knowledge and the episodic memory system to activate specific instances that exemplify, contextualize and ground their underlying themes or concepts [90]. He also postulated that memories from adolescence and early adulthood, known as the “reminiscence bump”, have significant influence in supporting self-knowledge, notably because during this time a greater number of memories that are relevant to identity formation are encoded and remain highly accessible for later retrieval [99,100,101]. However, there is no evidence that memories are essential to the formation of conceptual self-knowledge, and other perspectives suggest that our conceptual self-knowledge may be “computed” at any given moment with relevant behavioral exemplars from autobiographical memory (e.g., remembering having long conversations makes me think I am talkative). Another possibility is that conceptual self-knowledge is independent of autobiographical event memories [94,95,102], which means that there would be no need to reference past events stored in autobiographical memory to access an abstracted mental representation of ourselves (e.g., I am talkative). This view is further supported by neuropsychological case studies showing that individuals with severe episodic memory impairments still retain accurate knowledge about their personality traits [103,104,105,106,107]. Certainly, one of the most famous cases is that of patient KC, who suffered severe retrograde and anterograde episodic memory loss following a motorcycle accident and, despite undergoing dramatic personality changes and having no memory for any events from his past, could reliably report his post-accident personality [107]. However, the hypothesis that conceptual self-knowledge and autobiographical event memories are independent is only relevant to a very specific form of conceptual self-knowledge: personality traits. Further studies, which examined other varieties of self-knowledge, such as social roles, physical attributes, values and preferences, suggested that episodic memory might play a role in maintaining these aspects of the conceptual self [92,108]. Concordant with the case of patient KC, these studies show that neither the formation nor the maintenance of abstract trait self-knowledge, such as personality traits, relies on episodic memory. First, having accurate knowledge of one’s personality traits does not depend on the ability to retrieve episodic memories of the events that led to the creation of self-belief. Second, it is possible to form new beliefs about one’s personality traits without the ability to encode new episodic memories [104].

As suggested by Conway’s theory and supported in Prebble’s model, semanticized autobiographical memory might play an important role in forming and maintaining the conceptual self, including knowledge at the level of life story, life chapters and general events [65,90,91]. Conceptual self-knowledge is therefore likely to be stored as part of the semantic memory system [94,102,109,110] but appears to recruit distinct neural regions from episodic and semantic autobiographical memory [110]. Faculties such as personality traits, skills and physical characteristics are reported to rely on medial cortical structures, such as the dorsomedial and ventromedial prefrontal cortices, anterior and posterior cingulate cortices, median parietal cortices, temporal poles and insulae [111,112,113,114,115,116,117,118,119,120,121,122]. In addition, studies on SC have shown that some subcortical regions are also involved, such as the brain stem, colliculi, periaqueductal gray, hypothalamus and hypophysis [89,123,124].

### 3.2. The Self Extended in Time, or Autobiographical Memories

The idea of the self extended in time pertains to autobiographical memory, which progressively develops over time. Prebble et al. support that the self extended in time is the continuity of the self across time, meaning that, despite change, one continues to be the same person now as in the past, and will continue to be the same person in the future [65]. They suggest the existence of two kinds of self extended in time: autobiographical episodic memory and autobiographical semantic memory. The episodic version of autobiographical memory is based on Tulving and colleagues’ princeps work on autonoetic consciousness [125,126,127]. Autonoetic consciousness refers to the capacity to mentally travel through time, relive past experiences in an extremely vivid way and project oneself and one’s forward behaviors into the imagined future, based on past experiences. For instance, “I remember that evening in December when I had dinner in the tallest building in Montreal with a panoramic view; I was sitting near the window at the back, in the right-hand corner of the restaurant, and it was freezing”. Tulving uses the term “remembering” to refer to autonoetic consciousness, which also encompasses the ability to project oneself into the future, based on one’s previous experience. “On my next trip, I would like to go back there, but this time I would rather sit in the middle of the restaurant, close to the fireplace”. Autonoetic consciousness is a defining property of episodic memory and a fundamental ability in the formation of our self-identity, which confers the feeling of being the same subject reliving a memory as the one who lived the event. Researchers suggest that autonoetic consciousness provides subjective experience phenomenological continuity which reminds us of the perceptions, thoughts and emotions that accompanied the original experience [65,77,126,127,128,129]. Therefore, episodic autobiographical memory allows one to build oneself as a constant evolving individual and plays a key role in the maintenance and the continuity of self. Nonetheless, the relationship between the present self and its extended-in-time equivalent is bilateral, as the “online” self makes autobiographical memory emerge. This might explain why we have no—or only very approximate—memories of early childhood: we would be able to encode knowledge that can then form autobiographical memories at about 24 months, when developing our “cognitive self” (i.e., SSS) as indexed by visual self-recognition [130,131].

The semantic version of autobiographical memory is supported by three different forms of semantic continuity [65]. The “semantic temporal chronology” would support a conceptual understanding of oneself as a temporally extended being and requires a preserved semanticized autobiographical memory. Semanticized autobiographical memory refers to memories initially strictly episodic or “specific” that have lost their contextual details and become “semanticized”; these are, for instance, summaries of repeated or extended events over time (e.g., “for the past decade, my friends and I have been getting together every year to go to the music festival ‘les Eurockéennes’. Each time, we camp at a different friend’s place and have barbecues that go on forever during the day, and then we go to the festival to listen to the concerts in the evening”). It may also play a vital role in facilitating a personalized chronology, including allowing one to imagine a personalized future, by providing the frame that allows episodic memory details to be recombined into novel future scenarios [132] (e.g., “This summer, the festival has been canceled because of the COVID-19 epidemic, but we are still able to get together for the weekend at a friend’s place, to camp and have barbecues, which, this time, will last all day and all night in a festive and authentic way, all with a musical backdrop”). The second form of semantic continuity is a “temporally extended SC”. The richness and sophistication of the SC make it resistant to change; thus, self-knowledge may provide a persistent model of self that would raise a sense of continuity, through the construction of a temporal chronology of our conceptual knowledge (e.g., “since I was a child, I’ve always loved music”) [71,91,109]. The conjunction of semantic temporal chronology and temporally extended self-knowledge leads to the third form of semantic continuity, namely the “narrative continuity” which is related to our life stories. The life story refers to the way we thematically and temporally organize information about our lives [133]. It is a narrow selection of autobiographical memories that collectively explain “how I came to be who I am” (e.g., when I was at University, I always got around by bike, and I realized it was a fast and ecological way to travel. Today, I only use my bike to go to work) [65,134]. Prebble’s vision of the self extended in time is highly elaborated, postulating several parallel mechanisms that allow episodic and semantic autobiographical memory to exist. Another exhaustive vision of autobiographical memory is Conway’s, which suggests that autobiographical memory arises from a combination of the episodic memory system and the long-term self [90]. Indeed, in the modified version of the SMS, the author proposes to integrate the long-term self, which implies the conceptual self that we cited above assimilated to the SC as it is classically depicted, and the autobiographical knowledge base. The autobiographical knowledge base itself consists of three broad areas: “lifetime periods”, which reflect overarching goals and activities (e.g., when I was at University, when I lived in Montreal); “general events”, which are categories of events linked across brief time periods (i.e., a week, a day, a few hours) or organized by a shared theme (first-time experiences, academic meetings); and “life-story schema”, which consists of an individual’s understanding of how the normative life story is constructed within our culture (e.g., my life as a woman in the 21st century, or my career as a hospital neuropsychologist).

Some other researchers have a more general conception of the self extended in time. Neisser describes it as “the self as it was in the past and as we expect to be in the future, known primarily on the basis of memory” [68]. Gallagher shares such a vision; in his view, the narrative self is extended in time to include memories of the past and intentions toward the future [69]. In the case of Damasio, the “extended consciousness” is the third and final layer of his model. He describes it as “a relatively stable collection of the unique facts that characterize a person, the ‘autobiographical self’” that depends on memories of past situations.

The anatomical substrates of both episodic and semantic autobiographical memory have been extensively explored in neuroimaging studies. The episodic form mainly relies on the brain networks that allow mental travel across time to retrieve memories, namely the hippocampus, which mediates recollection, and the posterior medial cortices, which support the visuospatial context processing of events, notably the temporo-parietal junction, the posterior cingulum, the precuneus and the retrosplenial cortex [92,135,136,137,138,139]. The semantic component of autobiographical memory notably involves the lateral temporal structures, and in particular the middle temporal gyrus, which is associated with general autobiographical knowledge related to personal events [140,141,142,143,144,145]. Finally, the insula would play a role in the foundation and primitive aspects of the self rather than in its extended conception. Based on this hypothesis, we suppose that damage to the insulae, and, consequently, eventual impairments of the most elementary components of the self, might therefore impact both episodic and semantic autobiographical memories.

## 4. Insula and the Self

The previous two sections reviewed the anatomical and functional aspects of the insula and depicted the self according to its main characteristics and through the eyes of different scientists. Here, we will mainly focus on the relationship between the insula and the present self. Indeed, only a few studies have been conducted so far on the self extended in time, i.e., autobiographical memory, in patients with insular damage, leaving only hypotheses regarding the relationship between the insula and the self extended in time.

### 4.1. Insula and the Subjective Sense of Self

As stated above, the IC is reported to be involved in processing a variety of bodily sensations, notably those related to interoception [40], leading to the assumption that the insula plays a role in generating core self-states. How is the insula involved in the SSS? Additionally, in what way does its impairment lead to an alteration in the SSS?

#### 4.1.1. Subjective Sense of Self in Healthy Insula

Very little work has focused on the link between the subjective experience and the insula. This is partly because of the difficulty in assessing the subjective experience. However, there is some evidence to suggest that the insular cortices are particularly involved in the SSS, from the deepest internal states of the body that we can perceive via interoception to metacognition capacities [1,49].

The assessment of interoceptive accuracy is performed by tasks assessing the individuals’ precision in monitoring their body activity, such as the heartbeat detection or heartbeat counting task [146,147]. For instance, a study conducted by Critchley demonstrated that the performance on interoceptive accuracy tasks was significantly and positively correlated with functional activation within the right IC and operculum, and that both the performance on the heartbeat detection task and the score on the subtest of the body perception questionnaire assessing awareness of bodily processes correlated with gray matter volume within the same regions [46]. Accordingly, a meta-analysis of neuroimaging studies found that interoceptive accuracy was associated with an increased activation of the IC, the somatosensory cortex, the precentral gyrus and the inferior frontal gyrus [148]. Moreover, a research team focused on non-painful gastric distention and showed that a subjective sense of fullness was associated with activation peaks in the bilateral dorsal PIC, the left MIC, the left AIC and the anterior cingulate cortex (ACC) [149]; such results therefore support the posterior to mid-anterior pattern of integration, from unconscious internal bodily states (i.e., stomach distention) to the interpretation of interoceptive sensations (i.e., satiety sensation). Another study revealed the involvement of the bilateral PIC and middle insulae when participants were aware of their own heartbeats, with right hemispheric dominance [148]. Beyond interoceptive accuracy, Critchley’s study found a strong relationship between negative affect and the blood-oxygen-level-dependent response in the right insula [46], thus supporting the proposal that the insula mediates interoceptive awareness and contains representation of bodily reactions in response to affective feeling states, in other words, somatic markers [48]. The higher-order somatosensory function, such as a subjective sense of body ownership, seems to be associated with the right IC. In an experiment using the rubber-hand illusion paradigm, Tsakiris and collaborators reported a positive correlation between activity in the right PIC and the sense of body ownership during the rubber-hand illusion, in which the subject was not actually moving but felt that the moving hand was their own [150]. However, Farrer and collaborators investigated the neural signatures of the sense of agency, using similar methods, such as the systematic manipulation of visual feedback to alter the experience of one’s body in action. They demonstrated that activity in the right PIC was correlated with the degree of matching between the performed and the viewed movement, and thus with self-attribution [151]. Beyond its involvement in the most primitive aspects of the SSS, the right insula seems to be implicated in more highly elaborated levels of awareness, such as visual self-recognition in comparison with the processing of another, highly familiar person’s face [152,153]. Moreover, in an experiment where subjects had to pass judgment between a self-referential condition versus a control condition in response to affectively normed pictures, the activation in medial areas, notably the bilateral insulae, was found in the condition requiring reflection on one’s emotional state, thus supporting a role in emotional awareness [154]; the authors even emphasized the role of medial areas in the integration of visceromotor aspects of emotional processing, with information gathered from the internal and external environments. Another study illustrating the relationship between interoceptive experience and emotional context highlighted correlations between activation, predominantly within the left dorsal ACC and bilateral AIC, and the intensity of negative context [155]. This implies that emotional states are integrated with interoceptive states in the representation of the subjective feeling of the moment.

#### 4.1.2. Subjective Sense of Self in Damaged Insula

As previously described, the insula receives data from all sensory modalities and has strong connections with both the limbic and autonomic systems, allowing one to create an awareness of one’s physical self now and across time. In other words, the insula by itself has a role in interoceptive awareness, namely what allows us to feel, understand and construe what is happening deep inside our bodies. Consequently, dysfunction of the insula is likely to lead to abnormal subjective feeling states and disrupt the SSS.

Meta-analyses of functional and structural studies are converging on a “common core” of areas that are affected across several psychiatric conditions [156,157]. Numerous studies reported significant gray matter decreases in the dorsal ACC and bilateral AIC in individuals with psychotic (i.e., schizophrenia) and nonpsychotic conditions (i.e., major depressive disorder, bipolar disorders, obsessive-compulsive disorder, substance use disorder and several anxiety disorders) [157]. For instance, a study found that patients remitted from anorexia nervosa showed reduced activation when hungry in the AIC and reduced functional connectivity between the right AIC and mid-dorsal insula and ventral caudal putamen, compared to healthy controls [158]. Such a disconnection could lead to failure in integrating taste information with homeostatic but also motivational drives. Interestingly, Naqvi et al. showed that insular damages diminished addictive behaviors, suggesting that it might be the result of a reduced ability to detect interoceptive states related to craving or a reduction in the hedonic feeling induced by the substances [159,160]. Concerning the involvement of the IC in psychotic conditions, it has been demonstrated that among populations at high risk for developing schizophrenia, subjects who go on to develop psychosis have decreased insular gray matter initially, compared to those who do not become psychotic [161,162]. Likewise, bilateral decreases in insular gray matter volumes are associated with schizophrenia with both positive and negative symptoms, possibly due to the misperception of the self as a distinct entity from the external world [163,164,165,166]. Furthermore, it has been demonstrated that lesions of the right PIC disturb the sense of limb ownership, leading to the sensation that a contralesional limb does not belong to one’s own body or even belongs to another person [167]. In addition to structural anomalies, the insula response during the processing of emotional facial expressions is abnormal in schizophrenia, and this process appears to involve functions subserved by the AIC, such as evaluating emotions, empathy and the theory of mind [168,169,170]. Thus, there seems to be a strong relationship between insular damage and mechanisms allowing one to distinguish what belongs to the self and what belongs to the non-self. These difficulties could reveal impairments at the level of the SSS. Neuropsychiatric symptoms reflecting perception deficits such as hallucinations and delusions have also been highlighted in patients with Alzheimer disease (AD), Parkinson’s disease (PD) and dementia with Lewy bodies (DLB), correlating with atrophy in the right PIC and, to a lesser extent, in the left AIC [171]. Moreover, interoceptive disorders have been demonstrated in neurological and developmental disorders. For instance, in patients with autism spectrum disorders (ASD), self-reported poor awareness of one’s own and others’ feelings were associated with a reduced response in the interoceptive IC [43,46]. In another study, Gracia-Cordero et al. investigated interoceptive awareness neural correlates within a cohort of patients with behavioral variant frontotemporal dementia (FTD), AD and frontal strokes, and control subjects. Participants were asked to tap a keyboard in time with their own heartbeat and then to estimate their confidence level at performing the task. Patients with AD and FTD showed a significant deficit in their confidence in reporting biological changes, which was related to atrophy across a broad frontotemporal, parietal and limbic-insular network [172]. PD is also known to be associated with reduced interoceptive accuracy and sensibility, which might even be taken as a proxy of insular degeneration in the disease, according to some researchers [1,173]. Another mechanism that may reveal impairments of the SSS is anosognosia. Philippi and collaborators published a case report about Henry, a patient with mild cognitive impairment due to AD who had disproportionate atrophy within the medial prefrontal cortex (MPFC) and the IC and presented unusual anosognosia [174]. In a study about memory awareness—a metacognitive function matching the self-awareness level in Pebble’s model—in AD, Cosentino and collaborators demonstrated a specific role for the right insula in supporting metamemory, with gray matter volume positively correlated with metamemory accuracy [175]. In line with this, a study in stroke patients revealed that the right IC was commonly damaged in patients with anosognosia for hemiplegia/hemiparesis but significantly less involved in the same population without anosognosia [32]. Other high-order awareness processing, such as the loss of subjective emotional awareness, or alexithymia, has been related to the degeneration of VENs in the AIC in FTD patients [176]. In the same line, a study found that high-functioning people with ASD displayed increased alexithymia which was correlated with reduced activation in the AIC [177], as well as brain-injured patients with pronounced AIC lesions being likely to acquire alexithymia [178]. Moreover, finally, in a cohort of patients with various neurodegenerative conditions (i.e., behavioral variant FTD, semantic dementia, progressive aphasia, progressive non-fluent aphasia and AD), bilateral insula integrity was found to be associated with social interaction insight [179].

In agreement with Craig’s work and pioneering experiments [6,35], numerous functional and structural neuroimaging studies have reported that the insula, and notably the right insula, seems to play a key role within several layers of the SSS, in both healthy and pathological subjects.

### 4.2. Insula and the Self-Concept

Self-related processing is known to engage several cortical regions, particularly along the midline, including the prefrontal cortex, posterior cingulate cortex and parietal regions [89]. Moreover, some neuroimaging studies have already reported insular activity, thus implying a potential link with the re-representations created by the anterior part of the insula, after analysis and interpretation of the internal bodily states that have been forwarded to its posterior part [111,180,181,182].

#### 4.2.1. Self-Concept in Healthy Insula

As mentioned earlier, the SC encompasses the body of autobiographical knowledge, self-esteem and self-image and refers to the prototype we have of ourselves. Therefore, it makes sense to consider that the insular region is involved in such processes, given the nature of its multiple connections with numerous brain regions, especially the ones associated with the limbic system, which is involved in the formation of memory and, consequently, of personal identity. Northoff et al. reviewed functional imaging studies of self-related tasks, namely tasks involving the judgment of personality traits, goals, abilities and physical appearance, which pertain to the SC. The authors emphasized the implication of the cortical midline structures, including the MPFC, the temporal poles and the IC [89]. In another study, Modinos and colleagues highlighted a major involvement of the left AIC, among other regions, and notably those associated with the limbic system, such as the MPFC and the anterior cingulate cortex, when subjects were engaged in self-reflection rather than when they reflected upon an acquaintance or general knowledge [111]. Interestingly, Kircher and collaborators found that the left side of the insula is involved in the content of self, such as self-knowledge of personality traits [153]. Another study about self-referential and social processing found a positive correlation between the degree of self-relatedness and activation in the left AIC—among other “core self” regions, such as the MPFC and the posterior cingulate cortex [183]. Interestingly, Perini and collaborators’ work about social salience of the self in adolescents found activations in the right AIC and dorsal ACC, when the evaluations were directed toward the self rather than others [184].

Thus, it is well-established that the AIC is a main actor in the representation of personality traits. Moreover, different patterns of activation in the insula would appear to be associated with different personality types, such as the novelty-seeking trait, which is correlated with a higher dopamine concentration in the right IC [185]. Furthermore, Johnson et al. found that an introvert personality was correlated with increased blood flow in the AIC, whereas an extravert personality was correlated with increased blood flow in the PIC [186].

#### 4.2.2. Self-Concept in Damaged Insula

Different affections such as psychiatric and neurologic disorders or acquired brain injury can affect the IC or interconnected structures, and therefore the SC.

A study conducted by Cicero et al. explored SC consistency, stability and clarity in patients with schizophrenia, a condition in which insular dysfunction is known to exist [156,157]. For this purpose, they used the Self-Concept Clarity Scale [187] and the Me-Not-Me Decision Task [188]. In the first task, participants had to rate statements (e.g., “My beliefs about myself often conflict with one another”) on a scale from “Strongly Agree” to “Strongly Disagree”; in the second task, participants had to decide whether or not 60 adjectives described themselves. Compared to healthy controls, the researchers observed that patients with schizophrenia, and notably patients with positive symptoms, had lower scores on the Self-Concept Clarity Scale. Moreover, they found more inconsistent responses to the Me-Not-Me Decision Task (e.g., responding “me” to both “shy” and “outgoing”) in participants with schizophrenia associated with negative symptoms [189].

A few studies have examined self-perception in other conditions associated with insular dysfunction, such as ASD [103,190]. In early research, Capps and collaborators explored SC in children with ASD versus typically developing children. They found lower global self-esteem and social competences in the ASD group compared to the typically developing children [191]. Another study on SC in adolescents with ASD showed that they perceive themselves to be less competent in a variety of domains, including social, athletic and peer likability domains, compared to typically developing adolescents [192].

Other interesting models to study the SC are neurodegenerative diseases. Indeed, AD is typically characterized by personality changes, and group studies have already emphasized a deterioration of conceptual self-knowledge, which is frequently reported by patients’ relatives [92,193]. In Henry’s case report, Philippi et al. reported that, in addition to the SSS disorders, the patient was unable to describe his SC, nor could he recall semantic autobiographical information, despite preserved general cognitive abilities [174]. Likewise, DLB patients also present difficulties in describing their SC [194]. In another study, researchers focused on the involvement of the IC in SC, and in particular its influence on personal tastes, in patients with early-stage DLB. They found that patients presented significant changes in tastes, in both food and non-food domains, compared to matched healthy control subjects. Moreover, these changes were negatively correlated to the bilateral IC volume [195]. Similar changes have already been observed secondary to isolated right insular infarct: patients presented changes in food and clothing tastes, among other behavioral (e.g., vegetative and sensory disorders) and cognitive symptoms (e.g., social, emotional and intuitive dysfunctions) [196].

Finally, there does not seem to be a consensus about the role of the different parts of the IC in the SC, although several studies have suggested a major involvement of the left insula when it comes to thinking of information about oneself [111,153,183].

### 4.3. Insula and the Self Extended in Time

Previous research has shown the insula to be a core region of the present-self networks. However, little is known about its contribution in autobiographical memory processing. The insula appears only as a secondary region of autobiographical memory in a healthy population [197]. However, autobiographical memory is impaired in psychiatric conditions and neurodevelopmental disorders associated with insular dysfunction, such as schizophrenia and ASD [198,199]. To our knowledge, there are no studies directly exploring the link between the insula and autobiographical memory. In line with Prebble’s model, and as supported by the case report of Henry who presented a severely impaired autobiographical memory, disproportionately for a prodromal stage of AD, we suggest that the dysfunction of autobiographical memory would be linked to a global collapse of the self, notably involving insular damage, through a breakdown in awareness, which has been proposed as a prerequisite for all other components of the self [65,174]. Given that the SSS and the SC are considered crucial for the formation, consolidation and maintenance of episodic and semantic autobiographical memory [65,68,98,125,131], we believe that impairments of the present self might lead, due to a cascade effect, to an alteration in the more highly elaborated components of the self, such as autobiographical memory. Future studies will need to explore the link between the components of the self extended in time and the insula. We suggest that DLB would constitute an interesting model to further support our hypothesis that insular damage engenders a global collapse of the self, with consequences for life memories. Indeed, we previously demonstrated that insular atrophy occurs bilaterally as early as the prodromal stage of DLB [200,201]. Moreover, we found that these early damages were also accompanied by a reduction in white matter volume in the brainstem [200], which is involved in the SSS [66]. Thus, we consider it likely that DLB patients would present a global impairment of the self, first related to the deterioration of the SSS.

## 5. Conclusions

This review has focused on the insula and its broad range of functions, the self through its different components and the relationship between the insula and the self. We have presented findings that indicate that the insula is involved in a wide variety of functions, ranging from sensory and affective processing to high-level cognition, such as processes constituting the self. Finally, scientists agree that the self exists in the present moment but also entails an extended-in-time version that refers to autobiographical memory. Moreover, they all agree on the existence of both a self as a subject and a self as an object. The link between the present self, notably as a subject, and the insula has been widely explored and is now well established. Our review found results supporting Craig’s hypothesis of a posterior-to-anterior insular axis of complexity [40]. Indeed, some studies support the notion that elementary aspects of the SSS, as assessed by interoception, body ownership or sense of agency, are sustained by the PIC [148,149,150,151,167], while more highly elaborated levels of the SSS, such as metacognitive abilities and emotional awareness, are sustained by the AIC [154,155,176]. Furthermore, as proposed by Craig’s work and pioneering experiments, the SSS seems to be specifically sustained by the right insula, thus suggesting that some aspects of the self could be lateralized [6,32,35,46,150,151,152,153,167,175]. Conversely, the involvement of the left insula is often reported in the literature when subjects are thinking of personal information about themselves [111,153,183], whereas the specific roles of the AIC and the PIC in the SC remain unclear. Finally, we are of the view that the insula’s contribution to autobiographical memory could be secondary to its strong involvement in the present self. We propose to conduct a holistic exploration of the different components of the self, from visceral brain to life memories. The most elementary aspects of the SSS could be studied by means of tasks assessing interoceptive awareness, such as the heartbeat detection task, and for the SSS processes of higher order, by means of questionnaires assessing anosognosia. The SC could be explored via questionnaires based on self-knowledge, evaluating the idea we have of who we are. Finally, autobiographical memory could be measured through the recollection of episodic and semantic memories. Scientifically, studying the self in DLB which is characterized by early insular atrophy would enable us to characterize the anatomic substrates and the relationships between the distinct components of the self. Clinically, such an exploration would provide a better understanding of the personality changes observed in the disease, with potential benefits for both the patients and their relatives. Finally, identifying the self-impairments within this population might lead to the development of self-focused cognitive remediation. Indeed, therapeutic research in neurodegenerative disease has not yet been successful for curative treatments. Thus, patients can mainly benefit from pharmacological treatments focusing on symptom alleviation. Hence, every way to improve cognition and quality of life for patients and their relatives must be considered, particularly non-pharmacological treatments. Therapies such as mindfulness-based cognitive therapy might help to improve elementary aspects of the self such as the SSS and consequently act upon higher-order self-components, such as autobiographical memory. Furthermore, recent studies in rodents demonstrated how the IC is plastic [202,203,204,205,206], thus opening new perspectives of treatments, such as repetitive transcranial magnetic stimulation, that our team is currently experimenting in DLB to explore the insula’s different roles (STIMLEWY study). By identifying self-impairments and their anatomical substrates in DLB, we could use the same methods to improve self-awareness, and also more broadly global cognition, sensory abilities and even emotional states.

## Figures and Tables

**Figure 1 biology-12-00599-f001:**
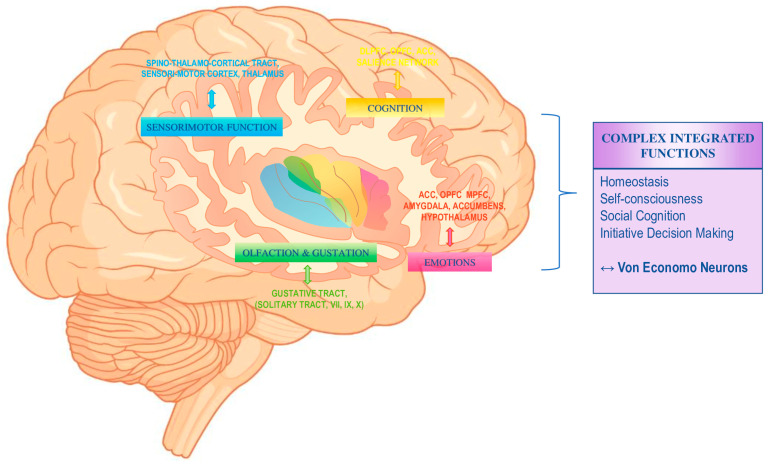
Functions and structural connectivity of the human insula. The blue section represents the sensorimotor function, the dark green section represents the gustatory function, the light green section represents the olfactory function, the yellow section represents the cognitive part, the red section represents the socio-emotional part and the purple section represents the complex integrated functions, stemming from the interaction of several of these functions. ACC, anterior cingulate cortex; DLPFC, dorsolateral prefrontal cortex; MPFC, medial prefrontal cortex; OPFC, orbitofrontal cortex. Adapted from “Icon Pack—Neuroscience”, by BioRender.com (2020). Retrieved from https://app.biorender.com/biorender-templates, accessed on 9 April 2023.

**Figure 2 biology-12-00599-f002:**
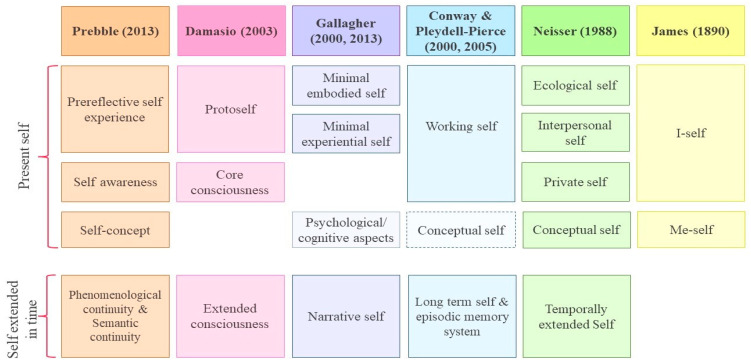
Synthesis of the main models of the self and autobiographical memory, with correspondences between the principal components. The orange frames depict Prebble’s model (2013), the pink frames depict Damasio’s model (2003), the purple frames depict Gallagher’s models (2000; 2013), the blue frames depict Conway and Pleydell-Pierce’s models (2000, 2005), the green frames depict Neisser’s model (1988) and the yellow frames depict James’s model (1890).

## Data Availability

Not applicable.

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
