# Peer review of "Me, Myself and My Insula: An Oasis in the Forefront of Self-Consciousness"

_biology, 2023, doi:10.3390/biology12040599_

Round 1
Reviewer 1 Report
Line 166:
“Within the PIC, the interoceptive pathway produces a topographical representation of the body from anterior to posterior aspects [4]”.
The article referenced is a review, while the statement could be misleading, as there is no direct evidence supporting this (in humans). Following up other related publications (e.g., Türe et al., 1999), where fixed brain tissue was studied, cannot be considered as evidence towards statement. The referenced (Line 170) study by Isnard et al. 2004 (Figure 1cf), would rather indicate that the feeling of unreality was reported by a small number of patients in response to stimulation of the inferior aspects of the AIC, MIC, and PIC.
Line 236:
Neisser, 1988 or [65]?
Line 385:
Self extended in time is developed gradually over time and refers to autobiographical memory.
I would encourage rephrasing this sentence. E.g.: The idea of self extended in time pertains to autobiographical memory, that progressively develops over time.
Line 683 on:
The authors have previously studied the role of defects at the IC in the prodromal stage of DLB - maybe relevant to discuss this further?
https://www.ncbi.nlm.nih.gov/pmc/articles/PMC5573371/
Conclusion:
Improving our understanding of these processes can indeed enhance our understanding of brain function relating to the SC and SSS in health and disease. How could these concepts be targeted in patient or risk populations?
The authors hint that cognitive therapy would be of benefit, but it would also be useful/informative to discuss this further. Of course, pharmacological treatments generally focus on symptom alleviation and the improvement of cognition, rather than specific brain regions or activity patterns. However, specifically relating to DLB, cholinesterase inhibitors, may rely on cholinergic activity in the insula to bring about their effects.
Further to this, considering the authors overall thesis, could TMS/rTMS regiments targeting the insula be of benefit?
Relevance/perspective from basic studies
Studies in rodents, would suggest that the topographical arrangement of representations within the insula are rather plastic (and multimodal) in relation to experience (Fletcher et al., Fontanini & Maffei et al., Rosenblum et al., Gogolla et al.), though others have argued for a quantitative mapping of sensory experiences (Zucker et al.). Recent evidence relating to the role of the aIC in encoding experiences over time, support several of the ideas put forward by the authors and may be of relevance to experimental manipulations of the SC and SSS (Gehrlach et al. 2019, Livneh et al. 2020, Vincis et al. 2020, Yiannakas et al. 2021, Koren et al. 2021, Djerdjaj et al. 2022, Schiff et al. 2023). The invasive nature of such basic studies (compared to what is possible in humans), could also provide insights into means of targeting the insula in patients in the future, the scope of effects that can be expected (cognitive, sensory, emotional but also immune states), but also the associated limitations and pitfalls.
Author Response
We are grateful for the precious time spent in reviewing our paper, for providing such valuable and insightful comments. Modifications in the manuscript have been highlighted in yellow.
Line 166:
“Within the PIC, the interoceptive pathway produces a topographical representation of the body from anterior to posterior aspects [4]”.
The article referenced is a review, while the statement could be misleading, as there is no direct evidence supporting this (in humans). Following up other related publications (e.g., Türe et al., 1999), where fixed brain tissue was studied, cannot be considered as evidence towards statement. The referenced (Line 170) study by Isnard et al. 2004 (Figure 1cf), would rather indicate that the feeling of unreality was reported by a small number of patients in response to stimulation of the inferior aspects of the AIC, MIC, and PIC.
Response : thank you very much for this comment that made us realize we switched two references. Thus, we corrected the reference “How do you feel now” Craig, 2009, by “Interoception : the sense of the physiological condition of the body” Craig, 2003, which highlights different findings suggesting a posterior-to-anterior pathway for interoceptive awareness in humans, we also specified that it is according to Craig’s theory. Moreover, we added a more recent study “Interoceptive awareness changes the posterior insula functional connectivity profile”, Kuehn et al., 2016, which supports Craig’s thesis (line 165). Moreover, we nuanced our statement concerning study by Isnard et al. 2004 (line 171).
Line 236:
Neisser, 1988 or [65]?
Response : thank you for this reminder that we have corrected. (line 238)
Line 385:
Self extended in time is developed gradually over time and refers to autobiographical memory.
I would encourage rephrasing this sentence. E.g.: The idea of self extended in time pertains to autobiographical memory, that progressively develops over time.
Response : thank you very much for this suggestion. We rephrased this sentence that is now much clearer. (line 387)
Line 683 on:
The authors have previously studied the role of defects at the IC in the prodromal stage of DLB - maybe relevant to discuss this further?
https://www.ncbi.nlm.nih.gov/pmc/articles/PMC5573371/
Response : thank you very much for this relevant remark. We incorporated our previous study in which findings support of our hypothesis concerning a global impairment of the self, due to deterioration of the SSS (line 715-720)
Conclusion:
Improving our understanding of these processes can indeed enhance our understanding of brain function relating to the SC and SSS in health and disease. How could these concepts be targeted in patient or risk populations?
Response : thank you for this question. We added a paragraph putting forward different ways to explore the different components of the self (line 744-750)
The authors hint that cognitive therapy would be of benefit, but it would also be useful/informative to discuss this further. Of course, pharmacological treatments generally focus on symptom alleviation and the improvement of cognition, rather than specific brain regions or activity patterns. However, specifically relating to DLB, cholinesterase inhibitors, may rely on cholinergic activity in the insula to bring about their effects.
Further to this, considering the authors overall thesis, could TMS/rTMS regiments targeting the insula be of benefit?
Response : thank you very much for this interesting suggestion. We added a paragraph to suggest different perspectives to overcome self impairments in neurodegenerative diseases, and more particularly in DLB (Line 765).
Relevance/perspective from basic studies
Studies in rodents, would suggest that the topographical arrangement of representations within the insula are rather plastic (and multimodal) in relation to experience (Fletcher et al., Fontanini & Maffei et al., Rosenblum et al., Gogolla et al.), though others have argued for a quantitative mapping of sensory experiences (Zucker et al.). Recent evidence relating to the role of the aIC in encoding experiences over time, support several of the ideas put forward by the authors and may be of relevance to experimental manipulations of the SC and SSS (Gehrlach et al. 2019, Livneh et al. 2020, Vincis et al. 2020, Yiannakas et al. 2021, Koren et al. 2021, Djerdjaj et al. 2022, Schiff et al. 2023). The invasive nature of such basic studies (compared to what is possible in humans), could also provide insights into means of targeting the insula in patients in the future, the scope of effects that can be expected (cognitive, sensory, emotional but also immune states), but also the associated limitations and pitfalls.
Response : thank you very much for these relevant references. In section 1.1., we added a sentence about IC multimodal organization in rodents (line 68). In section 4., we added some very interesting recent studies you suggested, to support new therapeutic perspectives (line755-768).

Reviewer 2 Report
This review focuses on the involvement of the insular cortex in numerous aspects of the self, and self impairments following insular damages. This is a very interesting review, well-written, and quite relevant.
Major comments:
Here are a few suggestions to improve the manuscript:
Comment #1. It could be relevant to add a paragraph about the insula, the self, and addiction. For example, Naqvi et al. (2007, 2009) showed that lesions to the insula diminished addictive behaviors, and this effect may be the result of a reduced ability to detect interoceptive states linked to craving or a reduction in hedonic feeling induced by the substances (section 3).
Comment #2. The literature provides evidence of interoceptive dysfunction in patients with eating disorders (e.g., hunger insensitivity in anorexia nervosa), and the involvement of insular cortex. A small explanation about this theme should be added (section 3).
Comment #3. In line 690, the authors mentioned a current limitation in the literature concerning the involvement of the insula in autobiographical memory (no data). Is there another topic that is not present in the literature, about the comprehension of the self and the insula (future investigation)? Maybe the phenomenon of alexithymia?
Comment #4. In line 722: “(…) the SSS seems to be specifically sustained by the right insula, thus suggesting that some aspects of the self could be lateralized”. Describe which other aspects.
Comment #5. Figure 1. I understand that figure 1 is an artistic rendition of the brain but the anatomy of the insula should be a little bit closer to reality. At present, we cannot identify the central insular sulcus, the two posterior long insular gyri nor the 3-4 short anterior insular gyri. I agree that the olfacto-gustatory area is in the middle insula but it is more dorsal than ventral. The words ‘spino-thalamo-cortical’ are underlined. Why is that?
Minor comments
There are some typographical errors. I mention here a few in the first 4 pages just to give some examples. A thorough review should be made for the whole manuscript:
Page 1 line 15 Simple summary: …and the way the insula…
Page 1 line 24 Abstract: …but also similarities
Page 1 line 26: awkward sentence; please rephrase.
Page 1 line 29; damages to the insular cortex
Page 2 line 52: opercula
Page 3 line 71: This is illustrated
Page 4 line 94: Heschl’s gyrus
Author Response
We are grateful for the precious time spent in reviewing our paper, for providing such valuable and insightful comments. Modifications in the manuscript have been highlighted in yellow.
Major comments :
Comment #1. It could be relevant to add a paragraph about the insula, the self, and addiction. For example, Naqvi et al. (2007, 2009) showed that lesions to the insula diminished addictive behaviors, and this effect may be the result of a reduced ability to detect interoceptive states linked to craving or a reduction in hedonic feeling induced by the substances (section 3).
Comment #2. The literature provides evidence of interoceptive dysfunction in patients with eating disorders (e.g., hunger insensitivity in anorexia nervosa), and the involvement of insular cortex. A small explanation about this theme should be added (section 3).
Response : thank you for these two very interesting suggestions that improved the part about SSS & insula in our manuscript. To introduce involvement of IC in SSS in psychiatric conditions, we added a paragraph about interoceptive dysfunction in patients with history of eating disorders and patients with addictive behaviors, line 557-565.
Comment #3. In line 690, the authors mentioned a current limitation in the literature concerning the involvement of the insula in autobiographical memory (no data). Is there another topic that is not present in the literature, about the comprehension of the self and the insula (future investigation)? Maybe the phenomenon of alexithymia?
Response : thank you for this relevant comment. Given that there is already a few data in literature about the involvement of insula in alexithymia, we chose to complete the paragraph in which we address the subject in section 3.1. (Line 607-610), rather than discussing it in the perspectives.
Comment #4. In line 722: “(…) the SSS seems to be specifically sustained by the right insula, thus suggesting that some aspects of the self could be lateralized”. Describe which other aspects.
Response : thank you very much for this reminder. We added a sentence after the one about implication of the right IC in SSS, to further discuss the involvement of left insula in the SC (Line 739-741).
Comment #5. Figure 1. I understand that figure 1 is an artistic rendition of the brain but the anatomy of the insula should be a little bit closer to reality. At present, we cannot identify the central insular sulcus, the two posterior long insular gyri nor the 3-4 short anterior insular gyri. I agree that the olfacto-gustatory area is in the middle insula but it is more dorsal than ventral. The words ‘spino-thalamo-cortical’ are underlined. Why is that?
Response : thank you very much for this comment. We modified the figure so that it is closer to reality from an anatomical point of view, both for the whole brain and for the insula. We tried to bring out the central insular sulcus, the two posterior long insular gyri and the short anterior gyri. Finally, we extended the gustatory part of the insula so that it is more on the dorsal side, and we modified the legend of the figure specifying that the gustatory part (which is more dorsal) is in dark green and the olfactory part (which is more ventral), is in light green.
Minor comments:
There are some typographical errors. I mention here a few in the first 4 pages just to give some examples. A thorough review should be made for the whole manuscript:
Response : thank you for having notice these errors, we carefully proofread the entire manuscript to correct them.

Round 2
Reviewer 2 Report
Authors have satisfactorily addressed our comments